# Deep Differentiable Logic Gate Networks

**Felix Petersen**
Stanford University
University of Konstanz
`mail@felix-petersen.de`

**Christian Borgelt**
University of Salzburg
`christian@borgelt.net`

**Hilde Kuehne**
University of Frankfurt
MIT-IBM Watson AI Lab
`kuehne@uni-frankfurt.de`

**Oliver Deussen**
University of Konstanz
`oliver.deussen@uni.kn`

## Abstract

Recently, research has increasingly focused on developing efficient neural network architectures. In this work, we explore logic gate networks for machine learning tasks by learning combinations of logic gates. These networks comprise logic gates such as "AND" and "XOR", which allow for very fast execution. The difficulty in learning logic gate networks is that they are conventionally non-differentiable and therefore do not allow training with gradient descent. Thus, to allow for effective training, we propose differentiable logic gate networks, an architecture that combines real-valued logics and a continuously parameterized relaxation of the network. The resulting discretized logic gate networks achieve fast inference speeds, e.g., beyond a million images of MNIST per second on a single CPU core.

## 1 Introduction

With the success of neural networks, there has also always been strong interest in research and industry in making the respective computations as fast and efficient as possible, especially at inference time. Various techniques have been proposed to solve this problem, including reduced computational precision [1], [2], binary [3] and sparse [4] neural nets. In this work, we want to train a different kind of architecture, which is well known in the domain of computer architectures: logic (gate) networks.

The problem in training networks of discrete components like logic gates, is that they are non-differentiable and therefore, conventionally, cannot be optimized via standard methods such as gradient descent [5]. One approach for this would be gradient-free optimization methods such as evolutionary training [6], [7], which works for small models, but becomes infeasible for larger ones.

In this work, we propose an approach for gradient-based training of logic gate networks (aka. arithmetic / algebraic circuits [8], [9]). Logic gate networks are based on binary logic gates, such as "and" and "xor" (see Table 1). For training logic gate networks, we continuously relax them to differentiable logic gate networks, which allows efficiently training them with gradient descent. For this, we use real-valued logic and learn which logic gate to use at each neuron. Specifically, for each neuron, we learn a probability distribution over logic gates. After training, the resulting network is discretized to a (hard) logic gate network by choosing the logic gate with the highest probability. As the (hard) logic gate network comprises logic gates only, it can be executed very fast. Additionally, as the logic gates are binary, every neuron / logic gate has only 2 inputs, and the networks are extremely sparse.

Logic gate networks are not binary neural networks: binary neural networks are a form of low precision (wrt. weights and/or activations) neural networks, as they reduce weights and/or activations to binary precision. Binary neural networks are usually dense and typically rely on weights trained in

36th Conference on Neural Information Processing Systems (NeurIPS 2022).

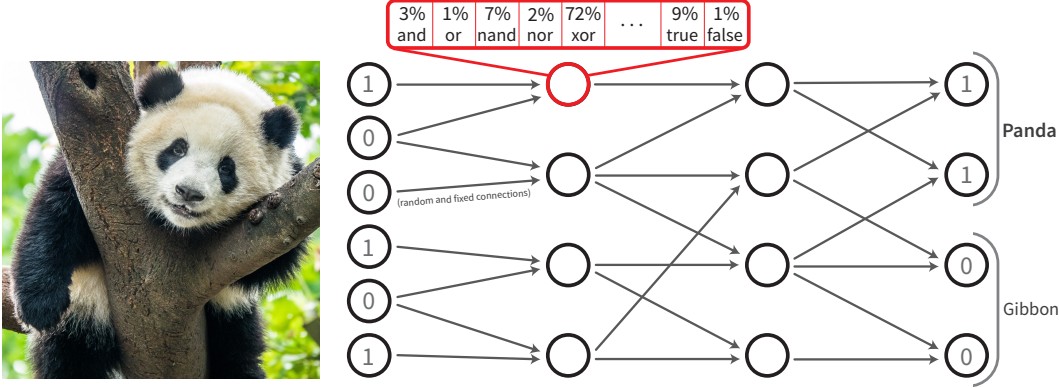

Figure 1: Overview of the proposed differentiable logic gate networks: the pixels of the image are converted into Boolean valued inputs, which are processed by a layer of neurons such that each neuron receives two inputs. The connectivity of neurons remains fixed after an initial pseudo-random initialization. Each neuron is continuously parameterized by a distribution over logical operators. During training, this distribution is learned for each neuron, and, during inference, the most likely operator is used for each neuron. There are multiple outputs per class, which are aggregated by bit-counting, which yields the class scores. The number of neurons in the visualization is greatly reduced for visual simplicity.

the continuous domain and are discretized afterwards. In contrast to binary neural networks, logic gate networks do not have weights, are intrinsically sparse as they have only 2 inputs to each neuron, and are not a form of low precision (wrt. weights and/or activations) neural networks.

They also differ from current sparse neural network approaches, as our goal is to learn which logic gate operators are present at each neuron, while the (weightless) connections between neurons are (pseudo-)randomly initialized and remain fixed. The network is, thus, parameterized by the choice of the logic gate operator / binary function for each neuron. As there is a total of 16 functions of signature $f : \{0, 1\} \times \{0, 1\} \rightarrow \{0, 1\}$, the information which operation a neuron executes can be encoded in just 4 bits. The objective is to learn which of those 16 operations is optimal for each neuron. Specifically, for each neuron, we learn a probability distribution over possible logic gates, which we parameterize via softmax. We find that this approach allows learning logic gate networks very effectively via gradient descent.

Logic gate networks allow for very fast classification, with speeds beyond a million images per second on a single CPU core (for MNIST at $> 97.5\%$ accuracy). The computational cost of a layer with $n$ neurons is $\Theta(n)$ with very small constants (as only logic gates of Booleans are required), while, in comparison, a fully connected layer (with $m$ input neurons) requires $\Theta(n \cdot m)$ computations with significantly larger constants (as it requires floating-point arithmetic). While the training can be more expensive than for regular neural networks (however, just by a constant and asymptotically less expensive), to our knowledge, the proposed method is the fastest available architecture at inference time. Overall, our method accelerates inference speed (in comparison to fully connected ReLU neural networks) by around two orders of magnitude. In the experiments, we scale the training of logic gate networks up to 5 million parameters, which can be considered relatively small in comparison to other architectures. In comparison to the fastest neural networks at $98.4\%$ on MNIST, our method is more than $12\times$ faster than the best binary neural networks and $2 - 3$ orders of magnitude faster than the theoretical speed of sparse neural networks.

## 2   Related Work

In this section, we discuss related work on learning logic gate networks, methods with methodological or conceptual similarity, as well as other machine learning methods that are fast and that we compare ourselves to with respect to inference cost and speed.

**Differentiable Logics and Triangular Norms**   Differentiable logics (aka. real-valued logics, or infinite-valued logic) are well-known in the fields of fuzzy logics [10] and probabilistic metric spaces [11], [12]. In Supplementary Material D we give examples for T-norms and T-conorms. An additional reference for differentiable real-valued logics is Van *et al.* [13].

**Learning Logic Gate Networks**   Chatterjee [14] explored "memorization", a method for memorizing binary classification data sets with a network of binary lookup tables. The motivation for this is to explore principles of learning and memorization, as well as their trade-off and generalization capabilities. He constructs the networks of lookup tables by counting conditional frequencies of data points. We mention this work here because binary logic gates may be seen as the special case of 2-input lookup tables. That is, his method has some similarities to our resulting networks. However, as he memorizes the data set, while this leads to some generalization, this generalization is limited. In his experiments, he considers the binary classification task of distinguishing the combined classes '0'–'4' from the combined classes '5'–'9' of MNIST and achieves a test accuracy of 90%. Brudermueller *et al.* [15] propose a method where they train a neural network on a classification task and then translate it, first into random forests, and then into networks of AND-Inverter logic gates, i.e., networks based only on "and" and "not" logical gates. They evaluate their approach on the "gastrointestinal bleeding" and "veterans aging cohort study" data sets and argue for the verifiability and interpretability of small logical networks in patient care and clinical decision-making.

**Continuous Relaxations**   A popular approach for making discrete structures differentiable is continuous relaxation [16], [17]. In this work, we also use continuous relaxations; however, instead of relaxing a fixed discrete structure (e.g., an algorithm) [16]–[23], we continuously relax a discrete structure (a logic gate network) to optimize it.

**Relaxed Connectivity in Networks**   In this work, we relax *which logic operator* is applied at each node, while the connections are predefined. Zimmer *et al.* [24] propose differentiable logic machines for inductive logic programming. For this, they propose logic modules, which contain one level of logic and for which they predefine that the first half of operators are fuzzy "and"s and the second half are fuzzy "or"s. They relax *which nodes* are the inputs to the "and"s and "or"s of their logic modules. Similarly, Chen [25] proposes Gumbel-Max Equation Learner Networks, where he predefines a set of arithmetic operations in each layer and learns via Gumbel-Softmax [26], [27], *which outputs* of the previous layer should be used as inputs of a respective arithmetic operation. He uses this to learn symbolic expressions from data. While these works relax which nodes are connected to which nodes, this is fixed in our work, and we relax which operator is at which node.

**Evolutionary Learning of Networks**   Mocanu *et al.* [28] propose training neural networks with sparse evolutionary training inspired by network science. Their method evolves an initial sparse topology of two consecutive layers of neurons into a scale-free topology. On MNIST, they achieve (with $89\,797$ parameters) an accuracy of $98.74\%$. Gaier *et al.* [29] propose learning networks of operators such as ReLU, sin, inverse, absolute, step, and tanh using evolutionary strategies. Specifically, they use the population-based neuroevolution algorithm NEAT. They achieve learning those floating-point function-based networks and achieve an accuracy of $94.2\%$ on MNIST with a total of $1\,849$ connections.

**Learning of Decision Trees**   Zantedeschi *et al.* [30] propose to learn decision trees by quadratically relaxing the decision trees from mixed-integer programs that learn the discrete parameters of the tree (input traversal and node pruning). This allows them to differentiate in order to simultaneously learn the continuous parameters of splitting decisions. Logic gate-based trees are conceptually vastly different from decision trees: decision trees rely on splitting decisions instead of logical operations, and the tree structure of decision trees and logic gate-based trees are in the opposite directions [31]. Logic gate-based trees begin with a number of inputs (leafs) and apply logic gates to aggregate them to a binary value (root). Decision trees begin at the root and apply splitting decisions (for which they consider an external input) to decide between children, such that they end up at a leaf node corresponding to a value.

**Binary Neural Networks**   Binary neural networks (BNNs) [3] are conceptually very different from logic gate networks. For binary neural networks, "binary" refers to representing activations and weights of a neural network with binary states (e.g., $\{-1, +1\}$). This allows approximating the expensive matrix multiplication by faster XNOR and bitcount (popcount) operations. The logical operations involved in BNNs are not learned but instead predefined to approximate floating-point operations, and, as such, a regular weight-based neural network. This is not the case for logic gate networks, where we learn the logic operations, we do not approximate weight-based neural networks, and do not have weights. While BNNs are defined via their weights and not via their logic operations,

logic gate networks do not have weights and are purely defined via their logic operations. We include BNNs as baselines in our experiments because they achieve the best inference speed.

**Sparse Neural Networks**  Sparse neural networks [4] are neural networks where only a selected subset of connections is present, i.e., instead of fully-connected layers, the layers are *sparse*. In the literature of sparse neural networks, usually, the task is to distill a sparse neural network from a dense neural network and the choice of connections is important. However, there has also been work suggesting a high effectiveness of using randomized and fixed sparse connections [32]. For logic gate networks, which are sparse by definition, we follow these findings and use randomly initialized and fixed connections.

## 3   Logic Gate Networks

Logic gate networks are networks similar to neural networks where each neuron is represented by a binary logic gate like 'and', 'nand', and 'nor' and accordingly has only two inputs (instead of all neurons in the previous layer as it is the case in fully-connected neural networks). Given a binary vector as input, pairs of Boolean values are selected, binary logic gates are applied to them, and their output is then used as input for layers further downstream. Logic gate networks do not use weights. Instead, they are parameterized via the choice of logic gate at each neuron. In contrast to fully connected neural networks, binary logic gate networks are sparse because each neuron has only 2 instead of $n$ inputs, where $n$ is the number of neurons per layer. In logic gate networks, we do not need activation functions as they are intrinsically non-linear.

While it is possible to make a prediction simply with a single binary output or $k$ binary outputs for $k$ classes, this is not ideal. This is because in the crisp case, we only get 0s or 1s and no graded prediction, which would be necessary for a "greatest activation" classification scheme. By using multiple neurons per class and aggregating them by summation, even the crisp case allows for grading, with as many levels as there are neurons per class. Each of these neurons captures a different piece of evidence for a class, and this allows for more finely graded predictions. Figure 1 illustrates a small logic gate network. In the illustration, each node corresponds to a single logic operator. Note that the distribution over operators (red) is part of the differentiable relaxation discussed in the next section.

As logic gate networks build on bit-wise logic operations only, their execution is very efficient.

## 4   Differentiable Logic Gate Networks

Training binary logic gate networks is hard because they are not differentiable, and thus no gradient descent-based training is conventionally possible. Thus, we propose relaxing logic gate networks to differentiable logic gate networks to allow for gradient-based training.

**Differentiable Logics**  To make binary logic networks differentiable, we leverage the following relaxation. First, instead of hard binary activations / values $a \in \{0, 1\}$, we relax all values to probabilistic activations $a \in [0, 1]$. Second, we replace the logic gates by computing the expected value of the activation given probabilities of independent inputs $a_1$ and $a_2$. For example, the probability that two independent events with probabilities $a_1$ and $a_2$ both occur is $a_1 \cdot a_2$. These operators correspond to the probabilistic T-norm and T-conorm; we report the full set of relaxations corresponding to the probabilistic interpretation in Table 1. (In addition, we report alternative relaxations corresponding to alternative interpretations in Tables 10 and 11 in Supplementary Material D.)

Table 1: List of all real-valued binary logic ops.

| ID | Operator | real-valued | 00 | 01 | 10 | 11 |
|----|----------|-------------|----|----|----|----|
| 0  | False | $0$ | 0 | 0 | 0 | 0 |
| 1  | $A \wedge B$ | $A \cdot B$ | 0 | 0 | 0 | 1 |
| 2  | $\neg(A \Rightarrow B)$ | $A - AB$ | 0 | 0 | 1 | 0 |
| 3  | $A$ | $A$ | 0 | 0 | 1 | 1 |
| 4  | $\neg(A \Leftarrow B)$ | $B - AB$ | 0 | 1 | 0 | 0 |
| 5  | $B$ | $B$ | 0 | 1 | 0 | 1 |
| 6  | $A \oplus B$ | $A + B - 2AB$ | 0 | 1 | 1 | 0 |
| 7  | $A \vee B$ | $A + B - AB$ | 0 | 1 | 1 | 1 |
| 8  | $\neg(A \vee B)$ | $1 - (A + B - AB)$ | 1 | 0 | 0 | 0 |
| 9  | $\neg(A \oplus B)$ | $1 - (A + B - 2AB)$ | 1 | 0 | 0 | 1 |
| 10 | $\neg B$ | $1 - B$ | 1 | 0 | 1 | 0 |
| 11 | $A \Leftarrow B$ | $1 - B + AB$ | 1 | 0 | 1 | 1 |
| 12 | $\neg A$ | $1 - A$ | 1 | 1 | 0 | 0 |
| 13 | $A \Rightarrow B$ | $1 - A + AB$ | 1 | 1 | 0 | 1 |
| 14 | $\neg(A \wedge B)$ | $1 - AB$ | 1 | 1 | 1 | 0 |
| 15 | True | $1$ | 1 | 1 | 1 | 1 |

Accordingly, we define the activation of a neuron with the $i$th operator as

$$a' = f_i(a_1, a_2),\tag{1}$$

where $f_i$ is the $i$th real-valued operator corresponding to Table 1 and $a_1, a_2$ are the inputs to the neuron. There are also alternative real-valued logics like the Hamacher T-(co)norm, the relativistic Einstein sum, and the Łukasiewicz T-(co)norm. While, in this work, we use the probabilistic interpretation, we review an array of possible T-norms and T-conorms that could also be used in SM D.

**Differentiable Choice of Operator**   While real-valued logics allow differentiation, they do not allow training as the operators are not continuously parameterized and thus (under hard binary inputs) the activations in the network will always be $a \in \{0, 1\}$. Thus, we propose to represent the choice of *which* logic gate is present at each neuron by a categorical probability distribution. For this, we parameterize each neuron with 16 floats (i.e., $\boldsymbol{w} \in \mathbb{R}^{16}$), which, by softmax, map to the probability simplex (i.e., a categorical probability distribution such that all entries sum up to 1 and it has only non-negative values). That is, $\boldsymbol{p}_i = e^{\boldsymbol{w}_i} / (\sum_j e^{\boldsymbol{w}_j})$, and thus $\boldsymbol{p}$ lies in the probability simplex $\boldsymbol{p} \in \Delta^{15}$. During training, we evaluate for each neuron all 16 relaxed binary logic gates and use the categorical probability distribution to compute their weighted average. Thus, we define the activation $a'$ of a differentiable logic gate neuron as

$$a' = \sum_{i=0}^{15} \boldsymbol{p}_i \cdot f_i(a_1, a_2) = \sum_{i=0}^{15} \frac{e^{\boldsymbol{w}_i}}{\sum_j e^{\boldsymbol{w}_j}} \cdot f_i(a_1, a_2).\tag{2}$$

**Aggregation of Output Neurons**   Now, we may have $n$ output neurons $a_1, a_2, ..., a_n \in [0, 1]$, but we may want the logic gate network to only predict $k < n$ values of a larger range than $[0, 1]$. Further, we may want to be able to produce graded outputs. Thus, we can aggregate the outputs as

$$\hat{y}_i = \sum_{j=i \cdot n/k+1}^{(i+1) \cdot n/k} a_j \, / \, \tau + \beta\tag{3}$$

where $\tau$ is a normalization temperature and $\beta$ is an optional offset.

## 4.1   Training Considerations

**Training**   For learning, we randomly initialize the connections and the parameterization of each neuron. For the initial parameterization of each neuron, we draw elements of $\boldsymbol{w}$ independently from a standard normal distribution. In all reported experiments, we use the same number of neurons in each layer (except for the input) and between 4 and 8 layers, which we call straight network. We train all models with the Adam optimizer [33] at a constant learning rate of $0.01$.

**Discretization**   After training, during inference, we discretize the probability distributions by only taking their mode (i.e., their most likely value), and thus the network can be computed with Boolean values, which makes inference very fast. In practice, we observe that most neurons converge to one logic gate operation; therefore, the discretization step introduces only a small error, e.g., for MNIST, the gap is smaller than $0.1\%$. We note that all reported results are accuracies after discretization.

**Classification**   In the application of a classification learning setting with $k$ classes (e.g., 10) and $n$ output neurons (e.g., $1\,000$), we group the output into $k$ groups of size $n/k$ (e.g., 100). Then, we count the number of 1s which corresponds to the classification score such that the predicted class can be retrieved via the $\arg\max$ of the class scores. During differentiable training, we sum up the probabilities of the outputs in each group instead of counting the 1s, and we can train the model using a softmax cross-entropy classification loss. For a reference on choosing the hyperparameter $\tau$, see Supplementary Material A.1; the offset $\beta$ is not relevant for the classification setting, as $\arg\max$ is shift-invariant. A heuristic for choosing $\tau$ is that when increasing $n$, we have to reduce $\tau$. Empirically, when increasing $n$ by a factor of 10, $\tau$ should be decreased by a factor of about 2 to $\sqrt{10}$.

**Regression**   For regression learning, let us assume that we need to predict a $k$-dimensional output vector. Here, $\tau$ and $\beta$ play the role of an affine transformation to transform the range of possible

predictions from 0 to $n/k$ to an application specific and more suitable range. Here, the optional bias $\beta$ is important, e.g., if we want to predict values outside the range of $[0, n/k/\tau]$. In some cases, it is desirable to cover the entire range of real numbers, which may be achieved using a logit transform $\text{logit}(x) = \sigma^{-1}(x) = \log \frac{x}{1-x}$ in combination with $\tau = n/k, \beta = 0$. During differentiable training, we sum up the probabilities of the outputs in each group instead of counting the 1s, and we can train the model, e.g., using an MSE loss.

## 4.2 Remarks

**Boolean Vectorization via Larger Data Types** One important computational detail for inference time is that we do not use Boolean data types but instead use larger data types such as, e.g., int64 for a batch size of 64, and thus perform bit-wise logics on larger batches which significantly improves speed on current hardware. For int64, we batch 64 data points such that the $i$th Boolean value of the $j$th data point is the $j$th bit in the $i$th int64 integer. Thus, it is possible to compute on average around 250 binary logic gates on each core in each CPU clock cycle (i.e., per Hz) on a typical desktop / notebook computer. This is the case because modern CPUs execute many instructions per clock cycle even on a single core, and additionally (for Booleans) allow single-instruction multiple-data (SIMD) by batching bits from multiple data points into one integer (e.g., int64). Using advanced vector extensions (AVX), even larger speedups would be possible. On GPU, this computational speedup is also available in addition to typical GPU parallelization.

**Aggregation of Output Neurons via Binary Adders** In addition, during inference, we aggregate the output neurons directly using logic gate nets that make up respective adders, as writing all outputs to memory would constitute a bottleneck and aggregating them using logic gate networks is fast. Specifically, we construct adders that can add exactly one bit to a binary number from logic gates.

**Memory Considerations** Since we pseudo-randomly initialize the connections in binary logic gate networks, i.e., which are the two inputs for each neuron, we do not need to store the connections as they can be reproduced from a single seed. Thus, it suffices to store the 4-bit information which of the 16 logic gate operators is used for each neuron. Thus, the memory footprint of logic gate networks is drastically reduced in comparison to neural networks, binary neural networks, and sparse neural networks.

**Pruning the Model** An additional speedup for the inference of logic gate networks is available by pruning neurons that are not used, or by simplifying logical expressions. However, this requires storing the connections, posing a (minor) trade-off between memory and speed.

**Subset of Operators** We investigated reducing the set of operators; however, we found that, in all settings, the more expressive full set of 16 operators performed better. Nevertheless, a smaller set of operators could be a good trade-off for reducing the model size.

**Half Precision** We also investigated training with half precision (float16). In our experiments, half precision (in comparison to full precision) did not degrade training performance; nevertheless, all reported results were trained with full precision (float32).

**Optimizer** For training differentiable logic gate networks, we use the Adam optimizer [33] because it includes a normalization of the gradients with respect to their magnitude over past steps. We found that this greatly improves training compared to other optimizers like SGD or SGD with momentum, which can become ineffective for training deeper logic gate networks.

## 5 Current Limitations and Opportunities

**Expensive Training** A limitation of differentiable logic gate networks is their relatively higher training cost compared to (performance-wise) comparable conventional neural networks. The higher training cost is because multiple differentiable operators need to be evaluated for each neuron, and in their real-valued differentiable form, most of these operators require floating-point value multiplications. However, the practical computational cost can be reduced through improved implementations.

We note that, asymptotically, differentiable logic gate networks are cheaper to train compared to conventional neural networks due to their sparsity.

**Convolutions and Other Architectures**    Convolutional logic gate networks and other architectural components such as residual connections are interesting and important directions for future research.

**Edge Computing and Embedded Machine Learning**    We would like to emphasize that the current limitations to rather small architectures (compared to large deep learning architectures) does not need to be a limitation: For example, in edge computing and embedded machine learning [34]–[37], models are already limited to tiny architectures because they run, e.g., on mobile CPUs, microcontrollers, or IoT devices. In these cases, training cost is not a concern because it is done before deployment.

We also note that there are many other applications in industry where the training cost is negligible in comparison to the inference cost.

## 6    Experiments[1]

To empirically validate our method, we perform an array of experiments. We start with the three MONK data sets and continue to the Adult Census and Breast Cancer data sets. For each experiment, we compare our method to other methods in terms of model memory footprint, evaluation speed, and accuracy. To demonstrate that our method also performs well on image recognition, we benchmark it on the MNIST as well as the CIFAR-10 data sets. We benchmark speeds and computational complexity of our method in comparison to baselines, which we discuss in detail in Section B.

### 6.1    MONK's Problems

The MONK's problems [40] are 3 classic machine learning tasks that have been used to benchmark learning algorithms. They consist of 3 binary classification tasks on a data set with 6 attributes with $2 - 4$ possible values each. Correspondingly, the data points can be encoded as binary vectors of size 17. In Table 2, we show the performance of our method, a regular neural network, and a few of the original learning methods that have been benchmarked. We give the prediction speed for a single CPU thread, the number of parameters, and storage requirements.

Table 2: Results on the MONK data sets. The inference times are per data point for 1 CPU thread. Averaged over 10 runs. For Diff Logic Nets, # Parameters and Space vary between the MONK data sets as we use different architectures.

| Method | MONK-1 | MONK-2 | MONK-3 |
|---|---|---|---|
| Decision Tree Learner (ID3) [38] | 98.6% | 67.9% | 94.4% |
| Decision Tree Learner (C4.5) [39] | 100% | 70.4% | 100% |
| Rule Learner (CN2) [31] | 100% | 69.0% | 89.1% |
| Logistic Regression | 71.1% | 61.4% | 97.0% |
| Neural Network | 100% | 100% | 93.5% |
| Diff Logic Net (*ours*) | 100% | 90.9% | 97.7% |

| | # Parameters | Inf. Time | Space |
|---|---|---|---|
| Decision Tree Learner | $\approx 30$ | 49ns | $\approx 60$B |
| Logistic Regression | 20 | 68ns | 80B |
| Neural Network | 162 | 152ns | 648B |
| Diff Logic Net (*ours*) | 144 \| 72 \| 72 | 18ns | 72B \| 36B \| 36B |

On all three data sets, our method performs better than logistic regression and on MONK-3 (which is the data set with label noise) our method even outperforms the much larger neural network. For hyperparameter details, see Supplementary Material A.1.

### 6.2    Adult and Breast Cancer

For our second set of experiments, we consider the Adult Census [41] and the Breast Cancer data set [42]. We find that our method performs very similar to neural

Table 3: Results for the Adult and Breast Cancer data sets averaged over 10 runs.

| **Adult** | Acc. | # Param. | Infer. Time | Space |
|---|---|---|---|---|
| Decision Tree Learner | 79.5% | $\approx 50$ | 86ns | $\approx 130$B |
| Logistic Regression | 84.8% | 234 | 63ns | 936B |
| Neural Network | 84.9% | 3810 | 635ns | 15KB |
| Diff Logic Net (*ours*) | 84.8% | 1280 | 5.1ns | 640B |
| **Breast Cancer** | Acc. | # Param. | Infer. Time | Space |
| Decision Tree Learner | 71.9% | $\approx 100$ | 82ns | $\approx 230$B |
| Logistic Regression | 72.9% | 104 | 34ns | 416B |
| Neural Network | 75.3% | 434 | 130ns | 1.4KB |
| Diff Logic Net (*ours*) | 76.1% | 640 | 2.8ns | 320B |

---

[1]The source code will be publicly available at github.com/Felix-Petersen/difflogic.

networks and logistic regression on the Adult data set while achieving a much faster inference speed. On the Breast Cancer data set, our method achieves the best performance while still being the fastest model. We present the results in Table 3.

## 6.3 MNIST

For our comparison to the fastest neural networks, we start by considering MNIST [43]. The methods we compare ourselves to are also discussed in further detail in the baselines section. In comparison to the fastest method achieving at least $98.4\%$ on MNIST, which is FINN by Umuroglu *et al.* [44] (identified by Qin *et al.* [3]), our logic network achieves a better performance, while requiring less than $10\%$ of the number of binary operations. That is, our model is objectively more than $10\times$ cheaper to evaluate. When comparing real times, for an NVIDIA A6000 GPU, our model is $12\times$ faster than the model by Umuroglu *et al.* [44] on their specialized FPGA hardware, even though our model only achieves a $7\%$ utilization of the GPU. For the other BNNs, OPs have not been reported, but their inference speed is also substantially slower than FINN. When compared to the smallest sparse neural network, our model requires substantially fewer operations than each of the of baselines. Sparse function networks [29], which have been learned evolutionarily, achieve an accuracy of $94.2\%$. We provide an additional discussion of the results displayed in the Table 4 in Supplementary Material B.

Table 4: Results for MNIST, all of our results are averaged over 10 runs. Times (T.) are inference times per image, the GPU is an NVIDIA A6000, and the CPU is a single thread at 2.5 GHz. For our experiments, i.e., the top block, we use binarized MNIST.

| MNIST | Acc. | # Param. | Space | T. [CPU] | T. [GPU] | OPs | FLOPs |
|---|---|---|---|---|---|---|---|
| Linear Regression | 91.6% | 4 010 | 16KB | $3\mu s$ | 2.4ns | (4M) | 4K |
| Neural Network (*small*) | 97.92% | 118 282 | 462KB | $14\mu s$ | 12.4ns | (236M) | 236K |
| Neural Network | 98.40% | 22 609 930 | 86MB | 2.2ms | 819ns | (45G) | 45M |
| Diff Logic Net (*small*) | 97.69% | 48 000 | 23KB | 625ns | 6.3ns | 48K | — |
| Diff Logic Net | 98.47% | 384 000 | 188KB | $7\mu s$ | (50ns) | 384K | — |
| *Binary Neural Networks* | | | | | T. [FPGA] | | |
| FINN [44] | 98.40% | | | $(96\mu s)$ | 641ns | 5.28M | |
| BinaryEye [45] | 98.40% | | | | $50\mu s$ | | |
| ReBNet [46] | 98.29% | | | | $3\mu s$ | | |
| LowBitNN [47] | 99.2% | | | | $152\mu s$ | | |
| *Sparse Neural Networks* | | | | | Sparsity | | |
| Var. Dropout [48] | 98.08% | 4 000 | | | 98.5% | (8M) | 8K |
| $L_0$ regularization [49] | 98.6% | | | | 2/3 | (200M) | 200K |
| SET-MLP [28] | 98.74% | 89 797 | | | 96.8% | (180M) | 180K |
| Sparse Function Net [29] | 94.2% | $3 \times 1\,849$ | | | | | > 2K |

## 6.4 CIFAR-10

In addition to MNIST, we also benchmark our method on CIFAR-10 [50]. For CIFAR-10, we reduce the color-channel resolution of the CIFAR-10 images and employ a binary embedding: For a color-channel resolution of $4$ (the first three rows of Table 5), we use three binary values with the three thresholds $0.25, 0.5$, and $0.75$. For a color-channel resolution of 32 (the large models, i.e., the last three rows of the top block of Table 5), we use 31 binary values with thresholds $(i/32)_{i\in\{1..31\}}$. We do not apply data augmentation / dropout for our experiments, which could additionally improve performance. For all baselines, we copied the reported accuracies from the original source, and thus those results are with data augmentation and the original color-channel resolution, and may include dropout [51] as well as other techniques such as student-teacher learning with a convolutional teacher [52].

The results are displayed in Table 5. We find that our method outperforms neural networks in the first setting (color-channel resolution of $4$) by a small margin, while requiring less than $0.1\%$ of the memory footprint and (with a larger model) by a large margin while requiring less than $1\%$ of the memory footprint. In comparison to the best fully-connected neural network baselines, which are trained with various tricks such as student-teacher learning and retain the full color-channel resolution, our model does not achieve the same performance, while it is also much smaller and has

access to fewer data. With 1 million parameters, the student-teacher model [52] has a footprint that is about $64\%$ larger than the footprint of our largest model (*large*$\times 4$), and achieves an accuracy which is only $3.7\%$ better than ours. It is important to note that this models requires 2 million floating-point operations, while our model requires 5 million bit-wise logic operations (before pruning/optimization). On float-arithmetic hardware-accelerated integrated circuits (as current GPUs and many CPUs), the 2 million floating-point operations are around $100\times$ slower than 5 million bit-wise logic operations. On general purpose hardware (i.e., without float acceleration) the speed difference would be one order of magnitude larger, i.e., $1\,000\times$.

More competitive with respect to speed are sparse neural networks. In the final block of Table 5, we report the sparsest models for CIFAR-10. Note that two of the methods resulted in performances below $50\%$, and even those methods which achieve around $75\%$ accuracy require a significantly more expensive inference. ProbMask [53] with its 140 KFLOPs per image means that the model is (depending on hardware) $1 - 2$ orders of magnitude more expensive than our largest model. Also, note that two out of three ResNet32 based sparsification methods achieve only around $37\%$ accuracy.

The results in parentheses are estimated because compilation to binaries did not finish / for GPU the largest models could also not be compiled due to compiler limitations. This can be resolved if desired with moderate implementation effort, e.g., compiling the model directly to PTX (CUDA Assembly) without compiling via `gcc` and `nvcc`. The actual problem is that the compilers used by the implementation have a compile time that is quadratic in the number of lines of code / statements.

We provide an additional discussion of the results displayed in Table 5 in SM B.

Table 5: Results on CIFAR-10. Times (T.) are inference times per image, the GPU is an NVIDIA A6000, and the CPU is a single thread at 2.5 GHz. For our experiments, i.e., the top block, we use a color-channel resolution of 4 for the first 3 lines and a color-channel resolution of 32 for the *large* models. The other baselines were provided with the full resolution of 256 color-channel values. The numbers in parentheses are extrapolated / estimated.

| **CIFAR-10** | Acc. | # Param | Space | T. [CPU] | T. [GPU] | **OPs** | FLOPs |
|---|---|---|---|---|---|---|---|
| Neural Network (color-ch. res. = 4) | 50.79% | 12.6M | 48MB | 1.2ms | 370ns | (25G) | 25M |
| Diff Logic Net (*small*) | 51.27% | 48K | 24KB | $1.3\mu s$ | 19ns | 48K | — |
| Diff Logic Net (*medium*) | 57.39% | 512K | 250KB | $7.3\mu s$ | 29ns | 512K | — |
| Diff Logic Net (*large*) | 60.78% | 1.28M | 625KB | $(18\mu s)$ | (73ns) | 1.28M | — |
| Diff Logic Net (*large*$\times 2$) | 61.41% | 2.56M | 1.22MB | $(37\mu s)$ | (145ns) | 2.56M | — |
| Diff Logic Net (*large*$\times 4$) | 62.14% | 5.12M | 2.44MB | $(73\mu s)$ | (290ns) | 5.12M | — |
| *Best Fully-Connected Baselines*  (color-ch. res. = 256) | | | | | | | |
| Regularized SReLU NN [28] | 68.70% | 20.3M | 77MB | 1.9ms | 565ns | (40G) | 40M |
| Student-Teacher NN [52] | 65.8% | 1M | 4MB | $112\mu s$ | 243ns | (2G) | 2M |
| Student-Teacher NN [52] | 74.3% | 31.6M | 121MB | 2.9ms | 960ns | (63G) | 63M |
| *Sparse Neural Networks* | | | | | Sparsity | | |
| PBW (ResNet32) [54] | 38.64% | | | | 99.9% | (140M) | (140K) |
| MLPrune (ResNet32) [55] | 36.09% | | | | 99.9% | (140M) | (140K) |
| ProbMask (ResNet32) [53] | 76.87% | | | | 99.9% | (140M) | (140K) |
| SET-MLP [28] | 74.84% | 279K | 4.7MB | | 98.6% | (558M) | 558K |

## 6.5 Distribution of Logic Gates

To gain additional insight into learned logic gate networks, we consider histograms of operators present in each layer of a trained model. Specifically, we consider a 4 layer CIFAR-10 model with $12\,000$ neurons per layer in Figure 2.

We observe that, generally, the constant $0/1$ "operator" is learned to be used only very infrequently as it does not actually provide value to the model. Especially interesting is that it does not occur at all in the last layer. In the first layer, we observe a stronger presence of 'and', 'nand', 'or', and 'nor'. In the second and third layers, there are more '$A$', '$B$', '$\neg A$', and '$\neg B$'s, which can be seen as a residual / direct connection. This enables the network to model lower-order dependencies more efficiently by expressing it with fewer layers than the predefined number of layers. In the last layer, the most frequent operations are 'xor' and 'xnor', which can create conditional dependencies of activations of the previous layers. Interestingly, however, implications (e.g., $A \Rightarrow B$) are only infrequently used.

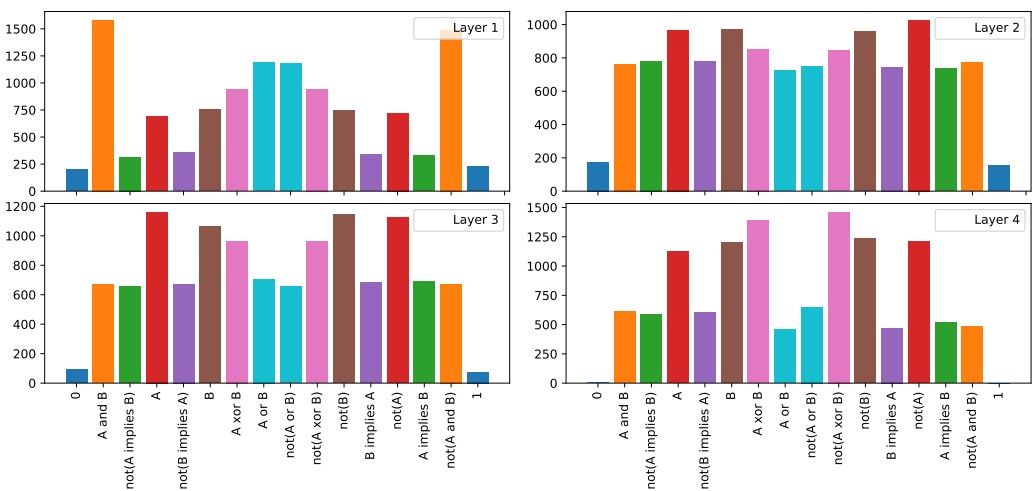

Figure 2: Distribution of logic gates in a trained four layer logic network.

# 7 Conclusion

In this work, we presented a novel approach to train logic gate networks, which allows us to effectively train extremely efficient neural networks that—for their level of accuracy—are one or more orders of magnitude more efficient than the state-of-the-art. For this, we leveraged real-valued logics and continuous relaxations via softmax. We will release the source code of this work to the community to foster future research on learning logic gate networks.

## Acknowledgments and Disclosure of Funding

This work was supported by the MIT-IBM Watson AI Lab, the DFG in the Cluster of Excellence EXC 2117 "Centre for the Advanced Study of Collective Behaviour" (Project-ID 390829875), and the Land Salzburg within the WISS 2025 project IDA-Lab (20102-F1901166-KZP and 20204-WISS/225/197-2019).

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
