# A Implementation Details

We will release the source code of this work in form of a library building on top of PyTorch [56] and include the presented experiments for reproducibility. For training, we use Python / PyTorch / CUDA, while for inference (apart from PyTorch) we use C for CPU and CUDA for GPU. The library supports both a pure PyTorch implementation as well as a native CUDA implementation (which is up to $50\times$) faster than the pure PyTorch implementation. Both implementations are usable in PyTorch: pure PyTorch / native CUDA just refers to the backend. For inference, our library automatically converts trained logic gate network into optimized C / CUDA binaries, allowing for easy and fast deployment, callable directly from Python (but also from any other language that can handle shared object binaries.)

## A.1 Model Architectures and Hyperparameters

For all experiments, we used "straight" architectures, i.e., architectures with the same number of neurons per layer. In Tables 6 and 7, we display the numbers of layers numbers of neurons per layer for each network architecture used in this work. In general, the architecture search for all models was performed via grid search with the number of layers in $\{2, 3, 4, 5, 6, 7, 8, 9, 10\}$ and number of neurons per layer with a resolution of factor 2.

For all models, we use the Adam optimizer [33]. For all neural networks, we use a learning rate of $0.001$ and for all logic gate networks, we use a learning rate of $0.01$. We train all models up to 200 epochs at a batch size of 100. The softmax temperature $\tau$ was searched over a grid of $\{1, 1/0.3, 1/0.1, 1/0.03, 1/0.01\}$ (except for Adult). As the experiments for were originally parameterized via an inverse temperature $1/\tau$, we provide the exact fractions to prevent rounding errors. The optimal temperature primarily depends on the number of outputs per class. If there are more outputs per class, the range of predictions is larger, and thus, we use a larger temperature to counter this effect.

Table 6: Logic gate network architectures.

| Dataset | Model | Layers | Neurons / layer | Total num. of p. | $\tau$ |
|---|---|---|---|---|---|
| MONK-1 | — | 6 | 24 | 144 | 1 |
| MONK-2 | — | 6 | 12 | 72 | 1 |
| MONK-3 | — | 6 | 12 | 72 | 1 |
| Adult | — | 5 | 256 | 1 280 | 1/0.075 |
| Breast Cancer | — | 5 | 128 | 640 | 1/0.1 |
| MNIST | small | 6 | 8 000 | 48 000 | 1/0.1 |
| | normal | 6 | 64 000 | 384 000 | 1/0.03 |
| CIFAR-10 | small | 4 | 12 000 | 48 000 | 1/0.03 |
| | medium | 4 | 128 000 | 512 000 | 1/0.01 |
| | large | 5 | 256 000 | 1 280 000 | 1/0.01 |
| | large×2 | 5 | 512 000 | 2 560 000 | 1/0.01 |
| | large×4 | 5 | 1 024 000 | 5 120 000 | 1/0.01 |

Table 7: Multi-layer perceptron / neural network baseline architectures. All architectures are ReLU activated.

| Dataset | Model | Layers | Neurons / layer | Total num. of parameters |
|---|---|---|---|---|
| MONK-1 | — | 2 | 8 | 162 |
| MONK-2 | — | 2 | 8 | 162 |
| MONK-3 | — | 2 | 8 | 162 |
| Adult | — | 2 | 32 | 3 810 |
| Breast Cancer | — | 2 | 8 | 434 |
| MNIST | small | 3 | 128 | 118 282 |
| | normal | 7 | 2 048 | 22 609 930 |
| CIFAR-10 | — | 5 | 1 024 | 12 597 258 |

### A.2  Aggregating Predictions via Logic Gate Network Adders

Optionally, we aggregate the output bits for each class into a binary number to reduce the required memory bandwidth for returning the predictions. This is done after learning and can be expressed via a fixed logic gate network. Specifically, we implement adders, which can add one bit to a binary number with logic gates. This way, the aggregation is extremely efficient, specifically, the aggregation is faster than storing the un-aggregated results in the VRAM.

### A.3  Training Times & Standard Deviations

Here, we provide training times for the MNIST and CIFAR-10 models. The training times are for the version of the code used for the original experiments. We will also make a substantially faster implementation publicly available. In addition, we provide the standard deviations for the accuracy.

Table 8: Training times and standard deviations for the experiments on MNIST.

| Model | Training Time | Accuracy |
|---|---|---|
| Neural Net Baseline (*small*) | 0.3 h | $97.92\% \pm 0.08\%$ |
| Neural Net Baseline | 0.4 h | $98.40\% \pm 0.06\%$ |
| Diff Logic Net (*small*) | 1.8 h | $97.69\% \pm 0.11\%$ |
| Diff Logic Net | 5.3 h | $98.47\% \pm 0.05\%$ |

Table 9: Training times and standard deviations for the experiments on CIFAR-10.

| Model | Training Time | Accuracy |
|---|---|---|
| Neural Net Baseline | 0.8 h | $50.79 \pm 0.35$ |
| Diff Logic Net (*small*) | 1.3 h | $51.27 \pm 0.26$ |
| Diff Logic Net (*medium*) | 7.4 h | $57.39 \pm 0.13$ |
| Diff Logic Net (*large*) | 24.2 h | $60.78 \pm 0.12$ |
| Diff Logic Net (*large*$\times 2$) | 45.6 h | $61.41 \pm 0.02$ |
| Diff Logic Net (*large*$\times 4$) | 90.3 h | $62.14 \pm 0.02$ |

## B  Additional Discussion of Baselines

In this section, we provide an additional discussion of fast network architectures as baselines for differentiable logic gate nets.

### B.1  Binary Neural Networks

Qin *et al.* [3] give a current overview of binary neural networks in their survey. They discuss the challenges of training binary neural networks or translating existing neural networks into their binarized counterparts. They identify FINN by Umuroglu *et al.* [44] as the fastest method for classifying MNIST at an accuracy of $98.4\%$ at a frame rate of $1\,561\,000$ images per second on specialized FPGA hardware.

FPGAs (field-programmable gate arrays) are configurable hardware accelerated processors that can achieve extreme speeds for fixed, predefined tasks that are expressed via logic gates. As FPGAs operate at extreme speeds, they were also used for applications such as mining cryptocurrencies [57], [58] or even implementing an oscilloscope [59] since, here, the required complexity is rather limited, while high speeds are necessary.

The binary FINN MNIST model by Umuroglu *et al.* [44] requires 5.82 MOPs (Mega binary OPerations) per frame, which means that their FPGA achieves around $5.82 \cdot 10^6 \cdot 1.561 \cdot 10^6 = 9.09 \cdot 10^{12}$ binary operations per second, i.e., 9.09 TOPS (Tera binary OPerations per Second). Conventional CPUs, are $10 - 100$ times slower than their FPGA. On different FPGAs, Ghasemzadeh *et al.* [46] achieve $330\,000$ images per second on MNIST at an accuracy of $98.29\%$, and Jokic *et al.* [45] propose an FPGA based embedded camera system achieving $20\,000$ images per second at $98.4\%$ accuracy.

Zhan *et al.* [47] concentrate on deploying Low-Bit Neural Networks (LBNNs) on FPGAs and achieve an accuracy of 99.2% on MNIST at 6 580 images per second. Shani *et al.* [60] explore analog logic gate nets (a physical approximation to Boolean nets) and achieve accuracies up to 89% on MNIST.

## B.2    Sparse Neural Networks

Sparse neural networks are neural networks where only a selected subset of connections is present, i.e., instead of fully-connected layers, the layers are *sparse*. Hoefler *et al.* [4] give an overview of sparsity on deep learning in their recent literature review. They identify Molchanov *et al.* [48] to achieve the sparsest (originally fully-connected) model on MNIST with a sparsity of 98.5% achieving an accuracy of 98.08%. For this, Molchanov *et al.* [48] propose variational dropout with unbounded dropout rates to sparsify neural networks. Louizos *et al.* [49] propose sparsification via $L_0$ regularization and report an MNIST accuracy of 98.6% for a model with around $2 \cdot 10^5$ FLOPs (FLoating point OPerations), which corresponds to a sparsity of around $2/3$.

Zhou *et al.* [53] propose ProbMask and give an overview over the sparsest CIFAR-10 models. They report up to a sparsity of 99.9%, which corresponds to around 140 kFLOPs for their smallest network architecture (ResNet32, which has a base cost of  140 MFLOPs [61].) This is the only work where architectures this sparse are reported in the literature. While these models have the advantage of being based on the VGG and ResNet CNN architectures, our models are still very competitive, especially considering that our models are much smaller than their smallest reported results.

Blalock *et al.* [62] report in their survey that for CIFAR-10 with a VGG, to achieve a theoretical speedup of $32\times$, all evaluated methods drop significantly below 70% test accuracy. Note that this speedup corresponds to a sparsity of around 97%, which makes up a much larger model than the models considered in this work.

Mocanu *et al.* [28] propose training neural networks with sparse evolutionary training inspired by network science. Their method evolves an initial sparse topology of two consecutive layers of neurons into a scale-free topology. They achieve an accuracy of 74.84% on CIFAR-10 with 278 630 floating-point parameters. We estimate this to correspond to a theoretical cost of around 550 kFLOPs (multiplication + addition), corresponding to 550 MOPs as per our conservative estimate. On MNIST, they achieve (with 89 797 parameters) an accuracy of 98.74%. Thus, the model is by orders of magnitude more expensive to evaluate than the logic gate networks considered in this work.

A FLOP generally corresponds to many binary OPs. Specifically, a float32 adders / multiplier requires usually at least 1 000 logical gates or look up tables and usually has a delay of tens of logical levels. Practically, float32 adders / multipliers are implemented directly in hardware in CPUs and GPUs, as it is an essential operation on such platforms. Nevertheless, also in practice, a float32 adder / multiplier is much more expensive than performing a bitwise logical operation on int64 data types (even on float32 and int32 focussed GPUs). On CPUs, around $3 - 10$ int64 bit-wise operations can be performed per cycle, while floating-point operations usually require a full clock cycle. To convert a non-sparse model we assume a very conservative 100 OPs per 1 FLOP. Note that speeds for sparse neural networks are also only theoretical because sparse execution usually brings an overhead of factor $10 - 100\times$. So overall, in practice, 1 000 (binary) OPs per 1 sparse (float32) FLOP is a very conservative estimate in favor of sparse float32 models. Further, in theory, 1 000 OPs per 1 FLOP is an accurate estimate (assuming sparsity to come without cost and assuming floating-point operations to not be hardware accelerated).

# C    Additional Discussions

## C.1    Depth vs. Accuracy Trade-off

We observed a trade-off between depth and accuracy that is similar to the trade-off for regular neural networks. In our experiments, we found that logic gate networks can generally be trained efficiently up to around 8-10 layers, when training starts to suffer from vanishing gradients. This is similar to where vanishing gradients start to be a problem in regular neural networks, at least without applying tricks like residual connections or batch norm.

## C.2 On the Effectiveness of Randomized Sparse Connections

We rely on fixed connections because learning the connectivity would require a relaxed connectivity, which would add additional complexity to the relaxation, which would degrade performance. However, we note that updating the connectivity based on some heuristic after a certain amount of training could, in principle, improve performance. This could be a subject of future work.

Sparsity can actually be viewed from two related angles: first, sparsity arises from the definition of binary logic gate operators leading to each neuron having only two inputs, which contrasts regular fully connected networks, where each neurons is a weighted sum of all inputs; second, sparsity can be seen from the perspective of the number of pairs of neurons covered: here, for $n$ inputs to a layer, we have $n \cdot (n+1)/2$ possible pairs of inputs, but typically only choose to consider $n$ pairs to avoid an exploding number of neurons in the downstream layers. Here, we use a random selection of connections as it is the canonical choice. We found that, as long as not only neighboring pairs of neurons are selected, the method of selection does not substantially affect performance, which is why we stuck with random connections to simplify the method. In future work, speed improvements (training and inference) could be possible by designing sparsity patterns that lead to faster memory access on respective hardware, while keeping the selection sufficiently "random" or "shuffled" such that accuracy is not impacted. As to why sparse and random connections work well in the first place, Liu *et al.* [32] discuss and investigate randomly selected sparse connections in regular neural network in great detail and demonstrate their effectiveness.

## C.3 Ternary and Other Additional Operators

In this work, we focus on binary logic gates. However, one may also consider ternary logic gates, i.e., logic gates with three inputs, e.g., $a \wedge b \wedge c$, or the more general form of $k$-ary logic gates. For $k$ binary inputs there are exactly $2^{(2^k)}$ possible binary operators. Thus, e.g., for 3 inputs, there are 256 possible binary operators. An important reason to limit the number of possible operators is that too many operators would lead to vanishing probabilities, thereby inhibiting training. Further, additional operators would lead to computationally more expensive training because more relaxed logic operators would need to be computed, more outputs would need to be aggregated, and more derivatives would need to be computed.

# D Differentiable Logics: T-Norms and T-Conorms

Here, we cover various T-norms and T-conorms, which are the build blocks of real-valued logics, and could be considered as alternatives to the probabilistic T-norm and T-conorm used in the main paper.

The axiomatic approach to multi-valued logics (which we need to combine the occlusions by different faces in a "soft" manner) is based on defining reasonable properties for truth functions. We state the axioms for multi-valued generalizations of the conjunction (logical "and"), called T-norms, in Definition 1 and generalizations of the disjunction (logical "or"), called T-conorms, in Definition 2.

**Definition 1** (T-norm). *A T-norm (triangular norm) is a binary operation $\top : [0,1] \times [0,1] \to [0,1]$, which satisfies*

- *associativity:* $\top(a, \top(b,c)) = \top(\top(a,b), c)$,
- *commutativity:* $\top(a,b) = \top(b,a)$,
- *monotonicity:* $(a \le c) \wedge (b \le d) \Rightarrow \top(a,b) \le \top(c,d)$,
- *1 is a neutral element:* $\top(a,1) = a$.

**Definition 2** (T-conorm). *A T-conorm is a binary operation $\bot : [0,1] \times [0,1] \to [0,1]$, which satisfies*

- *associativity:* $\bot(a, \bot(b,c)) = \bot(\bot(a,b), c)$,
- *commutativity:* $\bot(a,b) = \bot(b,a)$,
- *monotonicity:* $(a \le c) \wedge (b \le d) \Rightarrow \bot(a,b) \le \bot(c,d)$,
- *0 is a neutral element* $\bot(a,0) = a$.

**Remark 3** (T-conorms and T-norms). *While T-conorms $\bot$ are the real-valued equivalents of the logical 'or', so-called T-norms $\top$ are the real-valued equivalents of the logical 'and'. Certain T-conorms and T-norms are dual in the sense that one can derive one from the other using a complement (typically $1 - x$) and De Morgan's laws ($\top(a,b) = 1 - \bot(1-a, 1-b)$).*

Clearly, these axioms ensure that the corners of the unit square, that is, the value pairs considered in classical logic, are processed as with a standard conjunction: neutral element and commutativity imply that $(1,1) \mapsto 1$, $(0,1) \mapsto 0$, $(1,0) \mapsto 0$. From one of the latter two and monotonicity it follows $(0,0) \mapsto 0$. Analogously, the axioms of T-conorms ensure that the corners of the unit square are processed as with a standard disjunction. Actually, the axioms already fix the values not only at the corners, but on the boundaries of the unit square. Only inside the unit square (that is, for $(0,1)^2$) T-norms (as well as T-conorms) can differ.

In the theory of multi-valued logics, and especially in fuzzy logic [10], it was established that the largest possible T-norm is the minimum and the smallest possible T-conorm is the maximum: for any T-norm $\top$ it is $\top(a,b) \leq \min(a,b)$ and for any T-conorm $\bot$ it is $\bot(a,b) \geq \max(a,b)$. The other extremes, that is, the smallest possible T-norm and the largest possible T-conorm are the so-called drastic T-norm, defined as $\top^\circ(a,b) = 0$ for $(a,b) \in (0,1)^2$, and the drastic T-conorm, defined as $\bot^\circ(a,b) = 1$ for $(a,b) \in (0,1)^2$. Hence, it is $\top(a,b) \geq \top^\circ(a,b)$ for any T-norm $\top$ and $\bot(a,b) \leq \bot^\circ(a,b)$ for any T-conorm $\bot$. We do not consider the drastic T-conorm for an occlusion test because it clearly does not yield useful gradients.

As mentioned, it is common to combine a T-norm $\top$, a T-conorm $\bot$ and a negation $N$ (or complement, most commonly $N(a) = 1 - a$ so that DeMorgan's laws hold. Such a triplet is often called a *dual triplet*. In Tables 10 and 11 we show the formulas for the families of T-norms and T-conorms, respectively, which, together with the standard negation $N(a) = 1 - a$, form dual triplets.

Finally, we would like to recapitulate that, in this work, we used the probabilistic T-norm / T-conorm.

Table 10: (Families of) T-norms.

| Minimum | $\top^M(a,b) = \min(a,b)$ |
|---|---|
| Probabilistic | $\top^P(a,b) = ab$ |
| Einstein | $\top^E(a,b) = \frac{ab}{2-a-b+ab}$ |
| Hamacher | $\top_p^H(a,b) = \frac{ab}{p+(1-p)(a+b-ab)}$ |
| Frank | $\top_p^F(a,b) = \log_p\left(1 + \frac{(p^a-1)(p^b-1)}{p-1}\right)$ |
| Yager | $\top_p^Y(a,b) = \max\left(0, 1 - ((1-a)^p + (1-b)^p)^{\frac{1}{p}}\right)$ |
| Aczél-Alsina | $\top_p^A(a,b) = \exp\left(-(|\log(a)|^p + |\log(b)|^p)^{\frac{1}{p}}\right)$ |
| Dombi | $\top_p^D(a,b) = \left(1 + \left(\left(\frac{1-a}{a}\right)^p + \left(\frac{1-b}{b}\right)^p\right)^{\frac{1}{p}}\right)^{-1}$ |
| Schweizer-Sklar | $\top_p^S(a,b) = (a^p + b^p - 1)^{\frac{1}{p}}$ |

Table 11: (Families of) T-conorms.

| Maximum | $\bot^M(a,b) = \max(a,b)$ |
|---|---|
| Probabilistic | $\bot^P(a,b) = a + b - ab$ |
| Einstein | $\bot^E(a,b) = \bot_2^H(a,b) = \frac{a+b}{1+ab}$ |
| Hamacher | $\bot_p^H(a,b) = \frac{a+b+(p-2)ab}{1+(p-1)ab}$ |
| Frank | $\bot_p^F(a,b) = 1 - \log_p\left(1 + \frac{(p^{1-a}-1)(p^{1-b}-1)}{p-1}\right)$ |
| Yager | $\bot_p^Y(a,b) = \min\left(1, (a^p + b^p)^{\frac{1}{p}}\right)$ |
| Aczél-Alsina | $\bot_p^A(a,b) = 1 - \exp\left(-(|\log(1-a)|^p + |\log(1-b)|^p)^{\frac{1}{p}}\right)$ |
| Dombi | $\bot_p^D(a,b) = \left(1 + \left(\left(\frac{1-a}{a}\right)^p + \left(\frac{1-b}{b}\right)^p\right)^{-\frac{1}{p}}\right)^{-1}$ |
| Schweizer-Sklar | $\bot_p^S(a,b) = 1 - ((1-a)^p + (1-b)^p - 1)^{\frac{1}{p}}$ |