# OpenReview forum: "Deep Differentiable Logic Gate Networks"
_NeurIPS.cc/2022/Conference — NeurIPS 2022 Accept_

### Official Review · Reviewer_eqXk · 2022-07-11

**Rating:** 6
**Confidence:** 4
**Soundness:** 3 good
**Presentation:** 3 good
**Contribution:** 3 good

**Summary:**

This paper introduces a new approach for fast-inference, low memory neural networks: logic gate networks. An output neuron is connected sparsely to only two neurons from the input layer, and training learns one of sixteen possible boolean functions on the two binary inputs. Training is done with real relaxations of the logic functions and averaging over all 16 functions, while testing uses the most common function. Using logic gates enables extremely efficient inference execution on CPU. The authors show logic gate networks perform comparably to alternatives on several small datasets.

**Questions:**

Questions and suggestions:
- Line 195 I believe it should be "increasing n by a factor of 10, tau should be decreased..." not increasing tau.
- The paragraph in section 4.2 "aggregation of output neurons via binary adders" does not make any sense to me.
- I find the remark on lines 242-244 about being developed from scratch inappropriate for a paper. I would rewrite to discuss the possibility of future papers considering convolutions and other connectivity patterns without the excuse. And perhaps you should add a sentence explaining how the work is implemented (pytorch?)
- Higher training cost is mentioned for logic gate networks limitation in section 5. But how much higher is the cost? Quantifying this, even if the number might be improved down the line, seems like an important and useful benchmark to have.
- If you say practical cost of training can be reduced through improved implementations, then you should explain what stands to be improved and how much improvement we might gain.
- "Thus, our model is objectively more than 10x cheaper to evaluate" — I do not like this sentence. Whether a net is cheaper to evaluate depends on factors other than the number of binary operations.


**Limitations:**

Limitations:
- the limitations section is not very good. The statements are qualitative when it would be very easy to make them quantitative. Furthermore, there are more interesting things that could be said about future extensions. Most of the limitations section sounds like making excuses in case the reader is not impressed with the current work.

**Strengths And Weaknesses:**

Strengths:
- To my knowledge, this is a new idea. Using logic gates is a smart goal for achieving ultra-fast NN inference, and the authors seem to use the correct general approach to training such networks.
- Related work is well-cited, and the authors do a good job contextualizing to the space of similar techniques (sparse NNs/pruning, binarized NN's)
- Claims are fairly well supported.
- Submission is mostly well-written, with some exceptions.
- Significance: the results are not game-changing **yet**, but I find the goal of logic-gate networks to be an exciting direction and the work the authors achieve in pursuit of this direction.
- In all experiments, the approach presented has the fastest inference time.

Weaknesses:
- Overall writing quality could be better. Many details are not described well:
- The network architecture is poorly explained in section 3. The description is informal and at a high level of detail. The pieces don't really come together until the end of section 4; for example, the sparse net topology of 2 inputs per neuron is not fully explained with the random connectivity in line 221.
- There is no architecture figure. While not 100% necessary, I think this would be a helpful aid to understanding the architecture, especially how the connective topology is organized.
- Many training and architectural design choices are not motivated well. Often there are reasonable alternative choices, which should be ruled out, or better yet experimented against to see which is best! For example, there could be alternate ways of making the choice of logic gate differentiable (smoothly interpolate between logic ops?), or you could try structured connectivity patterns rather than random connectivity patterns. Or you could try expanding the set of operators rather than reducing. Four inputs instead of two?
- The paper has a lot of unpleasant language that seems to be making a lot of effort to apologize for poor results "if we had more time down the line, we could implement more connectivity types", or "practical computation cost can be reduced through improved implementations" and the like. Ignoring the quality of results, such statements do not belong in a paper in my opinion.
- Many of the baseline comparison architectures do not seem justified well. Why does the breast cancer dataset neural network have 3810 parameters? Why does the Diff logic net have 1280 parameters?
- It would be extremely valuable to have a comparison of training time for Diff Logic Nets and the baselines, even if some baselines' training time can't be given or can only be estimated. Even if DLN are much slower to train (that's okay!), this should be shown!

---

> ### Author Response · Authors · 2022-08-01
> **Author Response**
>
> Thank you for reviewing our paper and providing valuable feedback.
> Further, we want to thank you for the suggestions regrading the quality of writing.
> We agree with them and will incorporate all them, which will improve the clarity of the paper.
> Specifically, we will also explain the network architecture in more detail and add an architecture figure in the final paper.
>
> **Motivation of design choices.**
>
> Regarding structured connectivity patterns, in our paper, we explicitly chose to focus on random connections only as this is the vanilla variant of the method and does not induce additional inductive biases. We also considered different structured connectivity patterns but did not observe any improvement among those patterns that we considered.
> If only "neighbors" are connected, the model performance can be reduced.
> We attribute this observation to the fact that structured input can lead to the effect that only small local "neighborhoods" are considered.
> The randomization allows to circumvent those problems.
> To reduce the complexity of the method, we assume a random connectivity.
>
> Regarding expanded sets of operators, this is an interesting direction, but requires (1) consideration for differentiable relaxations of these operators and (2) will let the search space of operators explode as there are $2^{(2^k)}$ operators with $k$ inputs. For example, for four input, there would be $65536$ operators, and we would probably have to select a small subset of those to keep it feasible.
>
> Regarding the choice of logic gate operator, we are not aware of a direct alternative to the convex interpolation that we already perform. There could be alternative parameterizations for the interpolation, but softmax is not only robust but also easy to compute.
>
> As to the choice of differentiable relaxation of the logic gate operators, the probabilistic interpretation is the most natural. However, here we did some experiments with alternative T-norms and T-conorms (Tables 8 and 9) but among our tests, we did not find any choice that is better than the probabilistic variant.
>
> **Baseline comparison architectures**
>
> As to the baseline architectures, we performed a grid search over the number of layers and number of neurons per layer.
> For the number of neurons per layer, we chose a grid with a factor of 2. Concretely, for breast cancer and adult, we used the grid of {1, 2, 4, 8, 16, 32, 64, 128, 256, 512, 1024, 2048, 4096, 8192}.
>
> Baseline: For Adult, 2 layers with 32 neurons each performed best, and for breast cancer 2 layers with 8 neurons was best.
>
> DiffLogicNet: 5 layers of 256 neurons for adult, and 5 layers of 128 neurons for breast cancer.
>
> We note that the number of parameters for differentiable logic gate networks is smaller for the same number of neurons due to the sparsity.
> The number of parameters and the corresponding models can be found in Tables 6 and 7 in the supplementary.
>
> **Training time**: We provide training times for Table 4 and Table 5 in the general comment.
>
> **Binary Adders:**
> For the final classification, we consider the number of positive neurons within a certain output range (e.g. neuron 0-99 for the first class) and compare the amount to the amount in other ranges. For this comparison, we use a binary adder, which adds a single positive bit at a time. The resulting bit value can then be considered a discrete number that can be directly compared. Note that it would also be possible to assess the number of positive bits, e.g., by a counting loop, but that this solution would be inefficient. We will clarify this in the final paper.
>
>
> Thank you for the remark, we agree and we will clarify that training was done with PyTorch with custom extensions, and that the inference was done in C for CPU and CUDA for GPU.
> We will rewrite the limitations section, the discussion of future work, and remove the sentence as per your suggestion.
>
>
> Please let us know if you have any other questions regarding our paper or response. If we have successfully addressed your questions, we would highly appreciate an increased score.

---

> > ### Comment · Reviewer_eqXk · 2022-08-04
> > **Response**
> >
> > Thank you for the detailed response. I upgraded my score from a 5 to a 6.

---

### Official Review · Reviewer_2qAj · 2022-07-11

**Rating:** 8
**Confidence:** 4
**Soundness:** 3 good
**Presentation:** 3 good
**Contribution:** 3 good

**Summary:**

The authors proposed the Logic Gate Network - a neural network with neurons represented by binary logic gates. The network is parameterized by the choice of logic gates chosen from 16 logic operations. During training, the inputs to the neurons are relaxed from binary to probabilities to allow for differentiability. To allow for differentiability of the logic operations, each neuron has a weight vector of dimension 16, which when taken the softmax forms a categorical distribution. The activation of the neuron is the weighted sum of the distribution with their corresponding real-valued logic operation. At test time, the trained network is binarized to allow for fast inference. The authors evaluated their method on a set of classification tasks including MONK, Adult and breast cancer dataset, MNIST and CIFAR-10. Results show that compared to fully connected networks, the logic gate network is able to speed up inference time by 1-2 orders of magnitude and reduce the space consumption by roughly the same amount while moderately sacrificing accuracy in some tasks.

**Questions:**

- "after training, the network is binarized by choosing the logic gate with highest probability", this is not the same network that is being trained on, even though the binary network provides a speed up at inference, will the accuracy be higher with the "soft" version?

- "there is a total of 16 functions of signature" - can it be more than 16? if so will it improve performance?

- no comparison with CNN on image classification tasks, though this is for fair comparison with a fully connected logic net, can the authors comment how a CNN version of the logic net will perform?

- "CIFAR-10 reduced grayscale resolution and employ binary embedding",  why is this needed?

- The sparse neural network rows in Tables 4 and 5 are a bit confusing, they don't have inference times but instead sparsity metrics, which the diff logic net do not provide, not sure what the comparison here is.

**Limitations:**

The authors provided a reasonable set of limitations in Section 5. One of my concerns is that the input and output of the proposed logic network are Booleans, how would colored images be represented? Moreover, would regression tasks be out of the question for this type of network?

**Strengths And Weaknesses:**

Strengths
- The authors made good progress in enabling binarized logic network to work reasonably well in small to midsize classification problems while significantly reducing the memory footprint and speeding up the inference process. I particularly enjoyed Sections 4.1 and 4.2 where the authors discussed practical considerations and tricks they have tried to improve the performance. This type of work has much practical value to the edge deployment of neural network models.

Weaknesses
- elucidation of the position of the proposed architecture within the deep learning literature can be improved. functionality-wise, do the authors see the logic gated network able to achieve most of what other networks can down the road, or by design it would occupy a different functional space (for example efficient inference on the edge with reduced accuracy)?

---

> ### Author Response · Authors · 2022-08-01
> **Author Response**
>
> Thank you for reviewing our paper and providing valuable feedback. It helps us to improve the clarity of the paper.
>
> It is hard to make predictions regarding the future. In the near future, we think that the practical utility of the proposed architecture lies primarily in inference on the edge. But we are also optimistic that down the road logic gate networks could achieve most of what other networks can do now.
>
> **Discretization**:
> There is a small gap in accuracy due to discretization; however, we found that this gap is very small, e.g., for MNIST smaller than 0.1%.
>
> **Binary operators**:
> For $k$ binary inputs there are exactly $2^{(2^k)}$ possible binary operators. Thus, e.g., for 3 inputs, we would have 256 possible binary operators. However, constructing meaningful differentiable relaxations for these operators is not trivial and we estimate that having too many operators could result in vanishing gradients.
>
> **Convolution**:
> We currently do not have an implementation for a CNN version of logic gate networks, but we are optimistic and think this is an important direction.
>
> **Color images**:
> Currently, we separate each image into the three color channels and discretize the resulting channels separately in several steps leading to multiple binary "slices" for different thresholds for each image. A 32x32x3 CIFAR image that is binarized with 30 thresholds on each channel is thus represented by 3(RGB channels)*31(intervals induced by thresholds) 32x32 binary maps.
> We understand the potential for confusion and propose to change it from "grayscale resolution" to "color-channel resolution".
> Technically, it is also possible to forward the bits of RGB values directly. The problem here would be that lower bits would flip with much higher frequency, and thus producing rather random noise than a relevant signal, while the highest bits would rarely flip and therefore also encode less relevant information. This would be unfavorable for training and generalization.
>
> **Sparse NN baselines**:
> Regarding the sparse neural network baselines, the relevant metric for comparison is the number of operators (OPs).
> We agree that the comparison is not ideal, which is why we discuss their relations in the the experimental section, and propose to extend the discussion.
> The sparsity of differentiable logic gate networks is $1-\frac{2}{n}$ where $n$ is the number of neurons per layer.
>
> **Regression**:
> Regression tasks would also be possible with logic gate networks (though with a somewhat discretized output). We provide a short discussion in the last paragraph of Section 4.1.
>
>
> Please let us know if you have any other questions regarding our paper or response. If we have successfully addressed your questions, we would highly appreciate an increased score.

---

> > ### Comment · Reviewer_2qAj · 2022-08-07
> > **Response**
> >
> > I thank the authors for addressing my questions. Upgraded score to 8.

---

### Official Review · Reviewer_BrUk · 2022-07-15

**Rating:** 3
**Confidence:** 4
**Soundness:** 1 poor
**Presentation:** 1 poor
**Contribution:** 1 poor

**Summary:**

This paper proposes the differentiable logic gate networks. The logic gate networks parameterize the logical operations by proposing a logic gate neuron that consists of 16 binary logic operations. The differentiable learning is done by 1) using soft logic operations and 2) introducing attention parameters over the 16 operations. The proposed model is evaluated on standard classification tasks, which seems to suggest that the model is faster in inference time.

**Questions:**

I recommend the authors re-writing the paper and address the following concerns:
- What is the motivation of this work?
- What are the benefits of using this network compared to other neural-symbolic models?
- How are the layers connected? Is there any theoretical or empirical justification to this design?

I also recommend re-designing the experiments:
- Include training time when comparing with other baselines.
- Include empirical analysis on certain parts of the proposed network. For example, shifts in the feature space, and classification via grouping.


**Limitations:**

Yes

**Strengths And Weaknesses:**

### Originality
Overall, the proposed method isn't intellectually novel. The idea of using soft logic and attention mechanism to learn a logic system in a differentiable way has been a common practice in the literature, and similar and more sophisticated designs for the same problems have been proposed and studied extensively in many prior works [1-5].  With that being said, this paper clearly misses a big part of the literature. For those cited work, the paper does a poor job in connecting them to the proposed method, either due to the comparison is missing (e.g., L95) or the comparison is unconvincing and confusing:
- "while this leads to some generalization, this generalization is limited"
- "While these works relax which nodes are connected to which nodes,85this is fixed in our work, and we relax which operator is at which node"


### Quality
I find some of the proposed designs difficult to understand and can be fundamentally flawed:
- Judging from Figure 1 and section 4,  it looks like the neuron accepts only two inputs from the previous layer, but it is unclear how they are picked. This can be problematic compared to the standard fully connected layer because not all logical combinations are included in the model space. Consider this a simple setting with 3 binary inputs (a, b, c) and two neurons f1 and f2 that connect to (a, b) and (b, c) respectively. Then it is impossible to represent any operations between (a, c). To consider all combinations, one needs n^2 number of neurons at each layer.
- L187, the classification is performed by simply grouping adjacent neurons and counting. My concern is that this implicitly makes the output dependent on the position of the neurons. For the image classification task, this means the network is not translational invariant as opposed to CNNs, because a small shift in the original image is going to lead to different results.



### Clarity
The writing needs an overhaul. This paper is prone to confusing statements, constant grammatical errors and informal presentations.

The English writing should be improved:
- "as well as other machine learning methods that are fast and that we compare ourselves to with respect to inference cost and speed"
- Mixed usages of normal and italic "which" in L77-86

The wording should be improved; some of the contents are rather colloquial:
- "He does this to explore principles of learning and memorization" -> "The proposed method explores the  principles of learning and memorization"


### Significance
The proposed method is only evaluated on existing tasks and struggles to match with the performance of other baselines. The only merit seems to be that the proposed network excels in inference speed. However, this comes with serious limitations:
- As mentioned by the author at L235, the network is expensive to train. However, it's unclear how expensive the training actually is because only inference time is compared in the experiments. To fairly compare with other baselines, it is necessary to include the training time.
- The network likely oversimplifies the layer connections and can be flawed in representing certain logical combinations.

That said, there is no substantial contribution in this work.

[1] Serafini, Luciano, and Artur d'Avila Garcez. "Logic tensor networks: Deep learning and logical reasoning from data and knowledge." arXiv preprint arXiv:1606.04422 (2016).

[2] Dong, Honghua, et al. "Neural logic machines." arXiv preprint arXiv:1904.11694 (2019).

[3] Shi, Shaoyun, et al. "Neural logic reasoning." Proceedings of the 29th ACM International Conference on Information & Knowledge Management. 2020.

[4] Shanahan, Murray, et al. "An explicitly relational neural network architecture." International Conference on Machine Learning. PMLR, 2020.

[5] Yang, Yuan, and Le Song. "Learn to explain efficiently via neural logic inductive learning." arXiv preprint arXiv:1910.02481 (2019).

---

> ### Author Response · Authors · 2022-08-01
> **Author Response**
>
> Thank you for reviewing our paper, providing valuable feedback, and improving our paper.
>
> This work is not connected to neural-symbolic models nor any form of logical reasoning.
>
> Instead, the motivation of our work is purely to propose a new neural network architecture that is based on gates and is efficient in inference on typical hardware.
>
> Compared to neural-symbolic networks, which usually operate on higher-level concepts and features, the input of our system is the raw boolean data.
> E.g., in case of image classification, the input of the network are individual pixels reduced to boolean values via multiple thresholds. The input of a 32x32 CIFAR image is e.g. converted into a boolean 32x32x3x31 tensor (for 31 thresholds) and then flattened into a 95232 dim boolean vector.
>
> We could imagine that the abstract representation of the method in Figure 1 might be misleading here. We will clarify the overview to make it clearer and more comprehensive.
>
> We do not see any relation to neural-symbolic methods.
> Therefore, we compare in the related work mostly to low-precision / binary neural networks and sparse neural networks.
>
> We apologize for the misunderstanding and propose to add a section to the paper to clarify the difference between the two fields.
>
> Regarding convolution, we agree that for the image classification, the network is not translationally invariant because there is no convolutional structure that would allow for translational invariance. We will add a sentence about this to the paper.
>
> We provide training times for Table 4 and Table 5 in the general comment.
>
> We thank you for spotting the typos and for your suggestions regarding the quality of writing. We will implement them for the camera-ready version of the paper.
>
> Please let us know if you have any other questions regarding our paper or response. If we have successfully addressed your questions, we would highly appreciate an increased score.

---

> > ### Comment · Reviewer_BrUk · 2022-08-06
> > **Response**
> >
> > Thank you for the responses. I agree with the authors that this work differs from the neuro-symbolic methods in terms of the problem scope. Nevertheless, I do find the methodology applied here is very similar to that in the previous work.
> >
> > Thank you for providing the training time. It seems the proposed method is significantly more expensive to train, which probably would outweigh the gain obtained during inference time. If this is the case, the method really lacks significance in both performance and efficiency.
> >
> > While I appreciate the authors' effort in putting together the response, two of my biggest concerns still remain unresolved
> > - How are the two input neurons selected by the neurons in the next layer? How can you justify your proposed design? Right now it looks like they are fixed and this design seems flawed in this way.
> > - The final layer predicts by grouping adjacent neurons. This implicitly enforces a positional encoding on the neurons. Is this an intended behavior?

---

> > > ### Author Response · Authors · 2022-08-07
> > > **Response**
> > >
> > > Thank you for following up.
> > >
> > > > I do find the methodology applied here is very similar to that in the previous work.
> > >
> > > Could you please point us to the work that you find most similar in methodology compared to our work?
> > > For this, we would like to clarify that while we use softmax for the parameterization, we do not have an attention mechanism because the input to the softmax are parameters and we use softmax only to map from the 16 float parameters to the probability simplex. Attention would require that the inputs to softmax somehow depend on the input to the network or the activations, which is not the case in our work.
> > >
> > > Nevertheless, we agree that it is helpful to cover the works you referenced in the related work and discuss differences in methodology. Thus, we offer to include them in the the related work section of the final paper.
> > >
> > > > "Thank you for providing the training time. It seems the proposed method is significantly more expensive to train, which probably would outweigh the gain obtained during inference time. If this is the case, the method really lacks significance in both performance and efficiency."
> > >
> > > We respectfully disagree. Inference time matters a lot and should be considered independent of training time. For example, in industry, inference time is often substantially more important than training time as potentially tens (or even hundreds or thousands) of servers are deployed 24/7 to run a model. Reducing this cost is of substantial interest. Also in the domain of edge computing, where the limitation is the compute capability of a small processor / microcontroller, there is a large interest in reducing computational complexity of a model in order to be able to run it at all on a given hardware.
> > >
> > > Other methods that also have the goal of fast inference are developed and used even though their optimization is more computationally expensive than regular neural networks, e.g., binary neural networks or sparse neural networks.
> > >
> > > We would also like to mention that the provided training times were obtained using our original high-level PyTorch implementation. A low-level implementation for training logic gate networks is much faster because PyTorch itself does not have fast routines for differentiable logic gate networks.
> > >
> > > > How are the two input neurons selected by the neurons in the next layer? How can you justify your proposed design? Right now it looks like they are fixed and this design seems flawed in this way.
> > >
> > > The two input neurons for each neuron are selected randomly.
> > > We chose to use random connections as we consider this to be the vanilla variant of the method.
> > > Further, we found that, as long as we do not only select neighboring pairs of neurons, the method of selection did not affect performance, which is why we sticked with random connections to simplify the method.
> > >
> > > The connections remain fixed. This also avoids needing to learn the connectivity, which would require a relaxed connectivity, would add additional complexity to the relaxation, and thus degrade performance severely.
> > >
> > > For regular neural networks, fixed and randomly selected connections have been shown to perform very well (also compared to careful selection methods for the connections, and even compared to some dense neural networks) as shown by Liu et al. ("The Unreasonable Effectiveness of Random Pruning: Return of the Most Naive Baseline for Sparse Training", ICLR 2022, https://arxiv.org/pdf/2202.02643.pdf).
> > >
> > > If you would like to see a more extensive discussion, please see our comment "Re: Re: Response" to Reviewer r8L3.
> > >
> > > > The final layer predicts by grouping adjacent neurons. This implicitly enforces a positional encoding on the neurons. Is this an intended behavior?
> > >
> > > Yes, this is correct and an intended behavior. As we select the connections randomly, there is no correlation between the location of an output neuron and the location of an input (pixel). The same behavior also occurs in regular neural networks, e.g., the first output neuron may give the score for the class dog, while the output neuron at the second position may give the score for the class cat, and so on. In our case, we have to look at multiple neurons to obtain graded class scores, while regular neural networks only need one neuron per class as they already output real-valued activations.
> > >
> > > We will clarify all of these points in the camera-ready version of the paper.
> > >
> > > Please let us know whether we have successfully addressed your concerns and whether you have any other questions regarding our paper.

---

### Official Review · Reviewer_r8L3 · 2022-07-22

**Rating:** 8
**Confidence:** 4
**Soundness:** 4 excellent
**Presentation:** 4 excellent
**Contribution:** 4 excellent

**Summary:**

This paper proposes deep logic networks that can train with backpropagation and inference with binary values. The logic networks comprise logical gates arranged in layers, randomly selected connections between gates, and an output layer that sums the outputs from groups of gates to adapt cross-entropy loss or squared error loss. The types of gates are treated as parameters of the model and are optimized by Adam. The experiments show this network can perform classification tasks on MONK, Adult Census, Breast Cancer, MNIST, and CIFAR10, achieving good accuracy.

**Questions:**

1. Line 195: the first tau should be n?
2. Line 334: What is PTX?
3. What's the training time of each Diff Logic Net in Table 5?


**Limitations:**

The paper discussed 3 aspects of limitations in section 5.

**Strengths And Weaknesses:**

The paper is well written and easy to follow. To my best knowledge, the proposed Differentiable Logic Gate Networks is a highly novel idea. The model learns the type of gates instead of the weight matrix/connections. This training paradigm differs significantly from the standard DNN and other logic models. More interestingly, the Diff Logic Net uses randomly selected connections between gates, and the random connections are fixed during the training. This simple idea works surprisingly well and is probably the key reason it can learn the gate type in a scalable way. The experiments compare with a set of prior compact models and show the advantage of the Diff Logic Net in terms of accuracy, the number of operations, and actual inference time. I especially appreciate the results on CIFAR10 to demonstrate the scalability of this new breed of models. This work may significantly impact edge-device AI in both software and hardware aspects.

The success of Diff Logic Net looks too good to be true; therefore, discussing its limitation is highly appreciated. I saw it in Section 5 and consider the current section 5 reasonably informative. I believe the community will appreciate it more if section 5 is expanded further. One idea for exploring the current limitation is to examine its architectural robustness against several factors. For example:
1. Depth versus accuracy
2. If the input dimension is very high, would the random connection still work with a reasonable model width?
3. The variance of accuracy with different random connections

Note the above bullets are suggestions and I do not consider these are weaknesses of the paper.

---

> ### Author Response · Authors · 2022-08-01
> **Author Response**
>
> Thank you for reviewing our paper and providing valuable feedback and improving our paper.
>
> **Depth vs. accuracy**:
> We observed that the trade-off between depth and accuracy behaved similar to the trade-off for regular neural networks.
> In our experiments, we found that logic gate networks can generally be trained efficiently up to around 8-10 layers, where we observe that the training starts to suffer greatly from vanishing gradients.
> We note that this is similar to where vanishing gradients start to be a problem in regular neural networks without applying tricks like residual connections or batch norm.
> We will include a discussion of this in the final paper.
>
> **"If the input dimension is very high, would the random connection still work with a reasonable model width?"**:
> For this question, we would first like to note that the computational complexity of a fully-connected regular neural networks layer is quadratic in the number of neurons.
> At the same time, it is linear in the number of neurons for logic gate networks, as there are only 2 inputs per logic gate.
> Thus, a logic gate network with a width of $1\,000\,000$ is not comparable to a regular neural network of width $1\,000\,000$. Rather it is comparable to regular neural network of width $\approx 1\,000-10\,000$.
>
> In the case of CIFAR-10 with our large models, we had an input dimension of $32\times32\times3\times31=95\,232$; in this case, we used models with widths of $256\,000$, $512\,000$, and $1\,024\,000$. We assume that for networks with larger input dimension, we would need a larger width, but the width should remain reasonable.
>
> We also note that the necessary width primarily depends on the complexity of the data.
>
> **Variance of accuracy with different random connections**:
> This is an interesting point. Thus, we now provide the standard deviations for Tables 4 and 5 in the general comment. Here, we can observe that the variance is comparable to regular neural networks.
>
> **line 195**: Yes, it's a typo, we will fix it.
>
> **PTX**: PTX is an abbreviation for "Parallel Thread Execution". This is an assembly language / instruction set architecture by NVIDIA. We will clarify it in the revision.
>
> **training times**: We provide training times for Table 4 and Table 5 in the general comment.
>
>
> Please let us know if you have any other questions regarding our paper or response. If we have successfully addressed your questions, we would highly appreciate an increased score.

---

> > ### Comment · Reviewer_r8L3 · 2022-08-04
> > **Re: Response**
> >
> > Thanks for the detailed responses, and these make sense to me. Could the author share more sights or guess why fixed and sparse random connections work so well?

---

> > > ### Author Response · Authors · 2022-08-06
> > > **Re: Re: Response**
> > >
> > > We rely on *fixed* connections because learning the connectivity would require a relaxed connectivity, which would add additional complexity to the relaxation, which would degrade performance. However, we believe that updating the connectivity based on some heuristic after a certain amount of training could improve performance, which could be a subject of future work. *Sparsity* can actually be viewed from two related directions: first, sparsity arises from the definition of binary logic gate operators leading to each neuron having only two inputs, which contrasts regular fully connected networks, where each neurons is a weighted sum of all inputs; second, sparsity can be seen from the perspective of the number of pairs of neurons covered: here, for $n$ neurons as the input to a layer we have $n\cdot (n+1)/2$ possible pairs if inputs, but typically only choose to consider $n$ of them to avoid an exploding number of neurons in the deeper layers. Here, the *random* selection of the connections comes into play. We chose to use random connections as we consider this to be the vanilla variant of the method. We found that, as long as we do not only select neighboring pairs of neurons, the method of selection did not affect performance, which is why we sticked with random connections to simplify the method. We believe that, in future work, speed improvements (training and inference) could be possible by designing sparsity patterns that leads to faster memory access on respective hardware, while keeping the selection sufficiently "random" or "shuffled" such that accuracy is not impacted.
> > > As to why *sparse and random* connections work well in the first place, we recently found a work by Liu et al. ("The Unreasonable Effectiveness of Random Pruning: Return of the Most Naive Baseline for Sparse Training", ICLR 2022, https://arxiv.org/pdf/2202.02643.pdf), which discusses and investigates randomly selected sparse connections in regular neural network in great detail.

---

> > > > ### Comment · Reviewer_r8L3 · 2022-08-06
> > > > **Re: Re: Re: Response**
> > > >
> > > > The provided reference is very informative to the question. Your work and the reference jointly show that random sparse training could be a helpful strategy for creating compact models. These results are inspiring. Thank you!

---

### Author Response · Authors · 2022-08-01
**Training times and Standard deviations**

As requested by several reviewers, here, we provide training times for the models in Table 4 and Table 5.
In addition, we also provide the standard deviations for the accuracy results. We will also include these results in the final paper.

**MNIST**

| Model                             | Training Time | Accuracy [%]     |
|-----------------------------------|---------------|------------------|
| Neural Net Baseline (small)       | 0.3 h         | $97.92 \pm 0.08$ |
| Neural Net Baseline               | 0.4 h         | $98.4 \pm 0.06$  |
| Diff Logic Net (small)            | 1.8 h         | $97.69 \pm 0.11$ |
| Diff Logic Net                    | 5.3 h         | $98.47 \pm 0.05$ |

**CIFAR-10**

| Model                     | Training Time | Accuracy [%]     |
|---------------------------|---------------|------------------|
| Neural Net Baseline       | 0.8 h         | $50.79 \pm 0.35$ |
| Diff Logic Net (small)    | 1.3 h         | $51.27 \pm 0.26$ |
| Diff Logic Net (medium)   | 7.4 h         | $57.39 \pm 0.13$ |
| Diff Logic Net (large)    | 24.2 h        | $60.78 \pm 0.12$ |
| Diff Logic Net (large x2)    | 45.6 h        | $61.41 \pm 0.02$ |
| Diff Logic Net (large x4)    | 90.3 h        | $62.14 \pm 0.02$ |

---

### Meta-Review · Area_Chair_NhdK · 2022-08-26

**Recommendation:** Accept
**Confidence:** Less certain

**Metareview:**

The paper proposes to train classifiers for binary data based on logic gate networks, also called arithmetic circuits or algebraic circuits in the knowledge representation and complexity literature [1,2]. The main motivation is that, once trained, they can be run fast by leveraging only binary operations. The authors achieve this by using the classical way to relax logic operations, via a first-order approximation with fuzzy norms.

The reviewers found the proposed approach interesting and potentially useful for resource-constrained scenarios (where resources do not lack during training but during inference). I agree with them that this paper fits the tinyML literature more than the knowledge representation or reasoning (nowadays falling under the neuro-symbolic) one. However, I agree that believe the paper needs to fix certain aspects for the camera ready.

First, it is not clear why the authors are selecting the set of 16 logical operators, where, ideally only 2 are necessary (eg. AND, or product, and NOT, or negation) as all the others can be retrieved as combinations of the these two, as Table 1 illustrates. Furthermore, as using all 16 operators is the major bottleneck that makes the networks being very slow during training, an ablation test with a (even not minimal) subset of the operators is recommended. Clearly there is a trade-off in terms of accuracy w.r.t. the subset of ops used for a fixed depth network. And this trade-off needs to be made clear in the contribution (The authors can elaborate more than only saying "Omitting subsets of the operators decreased performance in all of our experiments.").

Second, there are a number of claims that need to be fixed. For example, the issue with differentiability with the relaxed functions is that gradients are constant everywhere, and only not defined on a single zero-measure point (between the zero and one regions). This is also one of the issues when trying to learn decision tree structures in a differentiable way. Additionally, the discussion on decision trees being very different from logic gate networks/algebraic circuits is very misleading and needs to be fixed. In fact, a decision node in a tree can be easily represented as a sum unit over some product units or equivalently OR and AND units in propositional logic or SMT [3, 4].

Third, since during training the authors are effectively using a probabilistic mixture model (Eqs 2,3), they are training something similar to a sum product network (without any relevant structural properties, and with negative weights) [5,1]. Therefore, the effect of the temperature parameter is important in retrieving a binary operator and not sufficiently investigated. An ablation study is recommended.

Lastly, for the larger experiments, it seems the authors are reporting the results of competitors which, however, use a different preprocessing (e.g. they use more gray-scale values for CIFAR). This makes the space and time consumption not comparable across models and requires a clarification.

I advice the authors to take care of these aspects to make the paper stronger.



[1] Darwiche and Marquis, "A Knowledge Compilation Map", 2001
[2] Arora and Barak, "Computational complexity: a modern approach" 2009.
[3] Khosravi, et al. "Handling missing data in decision trees: A probabilistic approach." 2020
[4] Devos, et al. "Verifying tree ensembles by reasoning about potential instances." 2021
[5] Choi, et al. "Probabilistic Circuits: A Unifying Framework for Tractable Probabilistic Models" 2020

**Award:**

No

---

### Decision · Program_Chairs · 2022-09-14

Accept